# Design of Photonic Crystal Biosensors for Cancer Cell Detection

**DOI:** 10.3390/mi14071478

**Published:** 2023-07-23

**Authors:** Yang Yang, Yang Xiang, Xubin Qi

**Affiliations:** School of Physics and Electronic Information Engineering, Henan Polytechnic University, Jiaozuo 454003, China; 212111010013@home.hpu.edu.cn (Y.Y.); 212111010011@home.hpu.edu.cn (X.Q.)

**Keywords:** silicon photonic crystal, resonant cavity, quality factor, sensitivity, cancer cells

## Abstract

A photonic crystal biosensor is a compact device fabricated from photonic crystal materials, which enables the detection and monitoring of the presence and concentration changes of biological molecules or chemical substances. In this paper, we propose a biosensor for cancer cell detection based on a silicon photonic crystal with a hexagonal resonant cavity introduced in a triangular lattice array. One of the bandgap ranges of this structure is 1188 nm≤λ≤1968 nm. When the incident light wavelength is within the range of 1188 nm≤λ≤1968 nm, the transmission coefficient of this structure at the resonant wavelength of 1469.58 nm is found to reach 99.62% through the finite difference time domain method, with a quality factor of 980. Subsequently, a biosensor is designed from this structure, with its sensing mechanism relying on the change in refractive index leading to a shift in the resonant wavelength. The target sample can be identified by observing the shift in the resonant wavelength. As cancer cells and normal cells possess different refractive indices, this biosensor can be used for their detection. The maximum sensitivity of the sensor is 915.75 nm/RIU and the minimum detection limit is 0.000236 RIU.

## 1. Introduction

According to data from the World Health Organization (WHO), approximately ten million people worldwide, died from cancer in 2020 [1]. Cancer is a disease characterized by irregular and uncontrolled cell growth that affects parts of the body. These abnormal cells, known as cancer cells, have the potential to lead to organ failure and mortality. However, if these cancer cells are detected and treated early, the survival rates, particularly for patients with breast cancer, will significantly increase [2]. At present, the main methods for early cancer diagnosis include regular physical examinations, blood tests, ultrasound imaging, pathological examinations, and cytological screenings. Although these methods can improve cure rates and survival rates for cancer, they exhibit some side effects. For instance, the utilization of imaging techniques such as computed tomography (CT) scans may result in a substantial amount of radiation, and excessive usage can heighten the risk of developing cancer. Positron emission tomography (PET) and magnetic resonance imaging (MRI) are not ideal for extensive utilization due to their high cost [3]. The lack of corresponding blood biomarkers in certain tumors decreases the sensitivity of blood tests for their detection. Therefore, researchers are focusing on the application of biosensors to detect diseases at an early stage.

Traditional biosensors make use of analytical techniques such as enzyme-linked immunosorbent assays, polymerase chain reactions, and fluorescence in situ hybridization for biological monitoring [4]. Currently, these biosensors require organic fluorescent dyes to detect cells. However, these dyes have poor photostability, broad absorption and emission ranges, and small Stokes shifts, making long-term or repeated analysis extremely complex. Therefore, the development of a rapid, efficient, sensitive, and label-free detection mechanism for cancer diagnostics is imperative in order to reduce the complexity and cost of clinical diagnosis. Photonic crystal (PC) biosensors have gained extensive research interest due to their characteristics, including high precision, miniaturization, high sensitivity, and the ability for label-free detection.

PC, an artificial periodic dielectric structure, was separately proposed by Yablonovitch [5] and John [6] in the 1980s. PC has two main characteristics: photonic bandgaps (PBG) and photonic localization. PBG refers to an energy range present in a photonic crystal where the transmission of specific wavelengths or frequencies of light is prohibited. The phenomenon of photonic localization appears when the periodic structure or symmetry of a PC is disrupted, engendering a narrow-bandwidth defect state within the PBG. This can substantially restrict the propagation of light and resist electromagnetic interference. As a result, they are often used as optoelectronic integrated circuits [7], phase controllers, PC optical field regulators, laser frequency stabilizers [8], and PC sensors [9,10,11,12].

Among these, PC sensors have attracted considerable attention globally due to their extensive utilization in the field of sensing. Common types of PC sensors include pressure sensors, liquid sensors, gas sensors, and biosensors. Vijaya Shanthi has proposed an L3 defect cavity pressure sensor based on a square array of silicon rods. By adjusting the radius of the L3 type defect, the sensor performance under pressures ranging from 0 to 7 GPa was analyzed. The sensor exhibits a sensitivity and dynamic range of 2 nm/GPa and 7 GPa, respectively [13]. Kassa-Baghdouche has conducted research on liquid sensors based on the H1-type and L3-type point defect nanocavities in elliptical hole planar photonic crystals (EPhC). Through optimization of the nanocavities, the liquid sensors achieved an ultra-high Q factor of 5000 and a sensitivity of 269 nm/RIU, respectively [14]. Kassa-Baghdouche has proposed the introduction of point defects in the center of a triangular lattice composed of circular air holes in a photonic crystal for use as a gas sensor. By adjusting the radius and position of the air holes in the defect region, the optimized sensitivity of the point-defect PC nanocavity was found to be 270 nm/RIU, with a detection limit of 10^−4^ RIU [15]. From the studies conducted by the aforementioned researchers, one can recognize the advantages of PCs in the field of sensing. Therefore, an increasing number of researchers are dedicating their efforts to the application of photonic crystals in the biomedical field, leading to the design of various types of biosensors.

Ineda demonstrated that photonic crystal (PC) biosensors provide a more efficient and convenient detection effect compared to scanning electron microscopy (SEM) and the α-casein chymotrypsin colorimetric method [16]. Parandin proposed a ring-shaped resonant cavity sensor, which determines the various components of blood by comparing the energy transmitted at the output. Although the sensor has a high quality factor of 5166, the sensitivity is only 2.94 nm/RIU [17]. Lower sensitivity may result in errors when detecting blood components with similar refractive indices. Krishnamoorthi proposed a square-embedded diamond-shaped biosensor for detecting various components in blood, with a maximum quality factor of 3702 and a sensitivity of 166 nm/RIU [18]. Olyaee proposed a laterally coupled ring resonator biosensor for detecting various components in blood, with a sensitivity of 272.43 nm/RIU and a quality factor of 3000 [19]. The biosensors designed by Krishnamoorthi and Olyaee have relatively high quality factor values and sensitivity which can mostly meet the requirements for detecting blood components. However, they are still insufficient to distinguish between normal cells and cancer cells. In recent years, researchers have designed highly sensitive biosensors.

Kiani designed a ring resonator sensor on a square grid of gallium arsenide rods for monitoring basal cells and normal cells, with a quality factor of 30 and sensitivity of 720 nm/RIU [20]. Jindal proposed a ring cavity biosensor that utilizes silica as the substrate and silicon as the dielectric rod to detect normal cells and cancer cells (in the range of 1.35 to 1.42), achieving a high sensitivity of 395 nm/RIU and a quality factor of 4800 [21]. Selfouri proposed a hexagonal resonant cavity biosensor based on a triangular lattice array for measuring water, ethanol, DNA, etc., with a maximum sensitivity of 561 nm/RIU and a quality factor of 3740 [22]. Fazea proposed a capsule-shaped sensor to detect glucose concentration in the human body, with a sensitivity of 546.72 nm/RIU and a quality factor of 2066.24 [23]. The capsule-type sensor designed by Fazea can effectively differentiate subtle changes in glucose concentration. Due to the high quality factor of the hexagonal resonant cavity, and the high sensitivity of the sensor which accurately reflects changes in glucose concentrations, we opted to integrate the structure of Selfouri’s hexagonal resonant cavity with Fazea’s capsule-like structure. To accommodate the detection of cancer cells, we have made some improvements based on this foundation, including the replacement of the outermost circular dielectric pillars of the hexagonal resonant cavity with four gear-shaped and eight ring shaped dielectric pillars. Additionally, the central dielectric pillar of the hexagonal resonant cavity was replaced with a capsule shaped pillar. In order to enhance the quality factor, we have added two semi-circular dielectric pillars to both sides of the capsule shaped pillar. In order to calculate the range of the photonic band gap (PBG), we employed the plane wave expansion (PWE) method and used the finite-difference time-domain (FDTD) method to simulate the characteristics of the sensor. By continually adjusting the radius of the dielectric pillars within the resonant cavity and optimizing parameters, we were able to design a high-quality sensor.

## 2. Sensor Performance Evaluation

Usually, the common parameters used to analyze sensors are the quality factor (*Q*), detection limit (*DL*), and sensitivity (*S*), as suggested in [24].

Quality factor is defined as the ratio between the resonant wavelength obtained (*λ*_0_) and the change in wavelength at full width half maximum (*FWHM*) (Δ*λ_FWHM_*), which can be expressed as [25]
(1)$$Q=\frac{\lambda_{0}}{\Delta \lambda_{FWHM}}$$

Sensitivity measures how much the output of the sensor varies with minute changes in input. It can be calculated by [26]
(2)$$S=\frac{\Delta \lambda }{\Delta n}$$
where Δλ is the position offset of the central wavelength in the transmission spectrum and Δn is changes in the refractive index.

*DL*, which stands for detection limit, is generally defined as the lowest amount or concentration of a component that can be detected using a given analytical method. The expression is
(3)$$DL=\frac{\lambda }{10QS}$$

The detection limit of a sensor entirely depends on the actual performance of sensitivity and quality factor indicators. The higher the sensitivity and the larger the quality factor, the lower the minimum amount or concentration of a component that can be detected.

## 3. Design and Performance Analysis of Biosensors

Biosensors based on the Silicon-On-Insulator (SOI) platform exhibit unparalleled advantages in various aspects encompassing manufacturing cost, space efficiency, and the quality of photonic devices. The exceptional compatibility with CMOS fabrication processes further amplifies their potential in the realm of modern microelectronic technology. The application of this technology enables the creation of miniature devices on a compact platform, characterized by high integration, superior performance, and low energy consumption. Silicon, a foundational material widely employed in the electronics industry, displays remarkable biocompatibility. This characteristic positions it as an advantageous choice for biosensor material selection, as it effectively reduces interference between the biosensor and biological samples, thereby enhancing the precision and stability of sensor detection. Hence, we selected silicon as the material for the biosensors.

Since the triangular lattice has a wider photonic bandgap than the square lattice, in design we adopt a triangular lattice to construct a complete PC structure consisting of 27 × 21 dielectric rods made of silicon in the *x*–*y* axis direction. The radius of the dielectric rod is r = 106 nm, and the distance between the dielectric rods is called the lattice constant, denoted by a, with a value of 630 nm. The dielectric constant of the circular silicon rod is 11.97 (n = 3.46), and the refractive index of the background is chosen as air (n_air_ = 1). The band diagram of the PC structure is obtained by using the plane wave expansion method, as shown in Figure 1.

From Figure 1, we can observe two Photonic Band Gaps (PBGs) represented by grey areas, with their PBGs at 0.32≤a/λ≤0.53 and 0.75≤a/λ≤0.81, corresponding to wavelength ranges of 1188 nm≤λ≤1968 nm and 777 nm≤λ≤840 nm, respectively. Here, ωa/2πc=a/λ is the normalized frequency with ω the angular frequency, and c the speed of light in vacuum. Since the first PBG is wide enough to cover the required wavelength range for optical communication applications, the wavelength of the optical source should be selected within the range of 1188 nm to 1968 nm.

In order to design the biosensor, we constructed resonant cavities as well as input and output waveguides by introducing line defects and point defects [27]. Point defects are created by removing a single block from the crystal or replacing it with another block of a different size or shape. Line defects are created by eliminating entire rows or columns of holes or rods. A straight waveguide is used for input, and a dropping waveguide for output. By introducing a point defect in the center of the structure, a hexagonal resonant cavity was constructed. Perfectly matched layer (PML) boundary conditions were set to allow the structure to fully absorb electromagnetic waves. Light was emitted towards the input end, which entered and coupled with the resonant cavity through the straight waveguide, and then transmitted to the monitoring device at the output end through the drop waveguide for observation of the transmission spectrum. The designed biosensor is shown in Figure 2. The dielectric pillars S_1_, S_2_, S_3_, and S_4_ all adopt a gear-like structure, inclusive of 20 external teeth, each with a height of 15 nm. The respective external radii are r_1_ = 0.25 μm, and r_2_ = r_3_ = r_4_ = 0.3 μm, while the internal radii are r_11_ = r_22_ = 0.15 μm and r_33_ = r_44_ = 0.08 μm. S_5_ is a combined medium column consisting of four semi-cylindrical medium columns and one rectangular medium column, with r_5_ of 0.2 μm and r_6_ of 0.1 μm. In the resonant cavity, the circular medium column has a radius of R = 0.18 μm, while the annular medium column has an outer radius of R_out_ = 0.18 μm and an inner radius of R_in_ = 0.12 μm. The amplification diagram of the resonant cavity is shown in Figure 3.

A Gaussian beam is injected through the input port along the straight waveguide, coupled via the hexagonal ring resonator. After propagation along the dropping waveguide, the transmission spectrum image is obtained as depicted in Figure 4 from the output port monitor.

As observed in Figure 4, at the resonant wavelength of 1455.29 nm, the transmission rate of the structure is recorded as 55.8% with a half-peak wavelength bandwidth of 2.7 nm, yielding a calculated quality factor of 537. Due to the low transmittance at the peak wavelength indicated in Figure 4, parameter adjustments are needed to further optimize the transmission rate and quality factor of this structure. The optimization process of these measurement parameters is discussed in Table 1. In the optimization process, firstly, we focus on optimizing the dielectric column S1 by adjusting the radius of r_1_, while keeping the radii of other dielectric columns unchanged. Considering the size of the S1 dielectric column in relation to the surrounding dielectric columns, we choose to increase the radius of r_1_ in increments of 0.05 μm. We find that when the radius of r_1_ is 0.3 μm, the monitor records a transmission rate of 97.6% and a calculated quality factor of 647. As the radius continues to increase, the resulting quality factor decreases. Therefore, we select r_1_ = 0.3 μm as the appropriate radius for the S_1_ dielectric column. Next, we optimize the S_2_ dielectric column while keeping r_1_ = 0.3 μm. We gradually increase the radius of r_2_ in steps of 0.05 μm and find that the best parameters are achieved when r_2_ = 0.32 μm. Keeping r_1_ = r_2_ = 0.3 μm, we continue to adjust the radius of the S_3_ dielectric column, r_3_, in steps of 0.05 μm. It is found that when the radius of r_3_ is 0.3 μm, the monitor records a transmission rate of 97.6% and a calculated quality factor of 647. However, when the radius of r_3_ is 0.35 μm, the monitor records a transmission rate of 98.74% and a calculated quality factor of 633. After considering the trade-offs, we choose the larger quality factor as the optimal parameter for the S_3_ dielectric column. Finally, when r_1_ = r_2_ = r_3_ = 0.3 μm, we adjust the radius of the S_4_ dielectric column by gradually increasing it in steps of 0.05 μm. We find that when the radii of r_1_, r_2_, and r_3_ are all 0.3 μm, and the radius of r_4_ is set to 0.4 μm, the transmission coefficient is 99.62% and the quality factor is 980, achieving optimal performance.

Figure 5 illustrates the optimal transmission spectrum obtained through optimizing the radii of the dielectric columns S_1_, S_2_, S_3_, and S_4_. As can be seen from Figure 5, the transmission rate at 1469.58 nm is close to 100%, with a computed quality factor of 980. Compared with Figure 4, a significant improvement in the transmission rate can be observed with an approximate increase of 44.2%. The quality factor also sees an increase of 443. This signifies that the sensor, post-optimization, possesses superior transmission performance. To verify whether the structure has reached its resonant state, a field monitor was used to measure the optical power distribution of the sensor. Figure 6 and Figure 7, respectively, depict the electric field distribution of the proposed sensor in OFF resonance and ON resonance conditions. As can be seen from Figure 6, at 1455 nm, which is in the OFF resonance state, almost no light enters the resonant cavity, with most of the energy concentrated in the straight waveguide. However, at 1469 nm, in the ON resonance state, light enters the resonant cavity for coupling, and energy is present in the dropping waveguide. These results demonstrate the feasibility of the PC biosensor.

Based on the previous analysis, when r_1_ = r_2_ = r_3_ = 0.3 μm and r_4_ = 0.4 μm in the structure, the quality factor of the structure is 980, and the transmittance is 99.62%, which is a substantial improvement compared to before optimization. Therefore, these parameters are chosen as the foundational conditions for the biosensor. Subsequently, the biosensor is fabricated by altering the refractive index of the resonant cavity dielectric columns. In this structure, we sequentially adjust the refractive index of the dielectric pillars, while keeping the others unchanged. The sensitivity and detection limit are calculated by measuring the resonant wavelength shift through numerical simulation.

The sensor is primarily utilized for detecting various cell types, including normal cells with a refractive index of 1.35, Jurkat cells with a refractive index of 1.39, PC12 cells with a refractive index of 1.395, MDA MB 231 cells with a refractive index of 1.399, and MCF 7 cells with a refractive index of 1.401 [28,29,30]. We sequentially investigate S_1_, S_2_, S_3_, and S_4_ dielectric pillars as the sensing sources of this structure, and the transmission spectra are monitored to plot the resonant wavelengths and quality factor. The sensitivity, detection limit, and quality factor of the biosensor are calculated utilizing the resonant wavelength of normal cells as a reference and are presented in Table 2.

As observed from Table 2, when using the S_1_ dielectric column as the detection source, the sensor achieves a maximum sensitivity of 915.75 nm/RIU and a minimum detection limit of 0.000236RIU, with a maximum wavelength shift of 39.89 nm. When the S_2_ dielectric column is used as the detection source, the obtained maximum sensitivity is 690.25 nm/RIU, the minimum detection limit is 0.000344RIU, and the maximum wavelength shift is 29.46 nm. When the S_3_ dielectric column is used as the detection source, the maximum sensitivity is 345.88 nm/RIU, the minimum detection limit is 0.000573RIU, and the maximum wavelength shift is 17.64 nm. When the S_4_ dielectric column is used as the detection source, the maximum sensitivity achieved is 609 nm/RIU, the minimum detection limit is 0.00036RIU, and the maximum wavelength shift is 26.68 nm. From Table 2, it is found that the sensor with the S_1_ dielectric column as the detection source has the best sensitivity, with a maximum wavelength shift of 39.89 nm. Therefore, this type of biosensor can effectively differentiate between normal cells and cancer cells. To further illustrate the role of the biosensor in distinguishing normal cells from cancer cells, Figure 8 displays the transmission spectrum with S_1_ as the detection source. As directly perceived from Figure 8, when the refractive index of the dielectric column changes, the wavelength experiences a shift.

Table 3 shows the comparison between the proposed biosensor and other sensors. Baratye proposed a biosensor based on an annular resonant cavity designed on a square lattice array for detecting normal cells, Jurkat cells, MDA MB 231 cells, and MCF 7 cells. The sensitivity and quality factor of the sensor were 308.5 nm/RIU and 3803.55, respectively [31]. Khan designed a ring resonator biosensor for detecting normal cells, Jurkat cells, MDA MB 231 cells, and MCF 7 cells, achieving a sensitivity of 227 nm/RIU, and a quality factor of 1200 [32]. Compared with the biosensor designed by Khan and Baraty, the biosensor designed in this paper demonstrates better sensitivity and transmission power and can accurately differentiate between cancer cells and normal cells. Asuvaran designed a diamond-connected-diamond resonant cavity biosensor for detecting gray matter cells, white blood cells, etc., with a maximum sensitivity of 4615 and a quality factor of 573 [33]. Bindal designed a square ring resonator biosensor for monitoring normal cells, Jurkat cells, MDA MB 231 cells, and MCF 7 cells, achieving a maximum sensitivity of 850 nm/RIU, and a quality factor of 650 [34]. Compared with the aforementioned sensors, the proposed structure achieved a relative balance between the sensitivity and the quality factor. As a result, it was able to effectively differentiate between normal cells and various cancer cells.

## 4. Conclusions

This paper innovatively combines the hexagonal resonator and the capsule-type resonator into a new structure, with adjustments made to the size and shape of the dielectric pillars. The twelve circular dielectric pillars on the outermost layer of the hexagonal resonator are replaced with four gear-shaped dielectric pillars and eight ring-shaped dielectric pillars. We introduced a capsule-shaped dielectric pillar in the center of the resonant cavity. By adjusting the radius of the dielectric pillars in the resonator, when r_1_ = r_2_ = r_3_ = 0.3 μm and r_4_ = 0.4 μm, the transmission efficiency of this structure reached 99.62% at 1469.58 nm, with a quality factor of 980. By adjusting the refractive index of the gear-shaped dielectric pillar in the resonant cavity, this structure can be used as a high-sensitivity biosensor capable of effectively distinguishing between cancer cells and normal cells. When pillar S_1_ is chosen as the detection source, the sensor reaches a maximum shift of 39.89 nm, a maximum sensitivity of 915.75 nm/RIU, and a minimum detection limit of only 0.000236. The biosensor proposed in this paper has high sensitivity and can be used for early detection of cancer cells as well as other biomedical applications.

## Figures and Tables

**Figure 1 micromachines-14-01478-f001:**
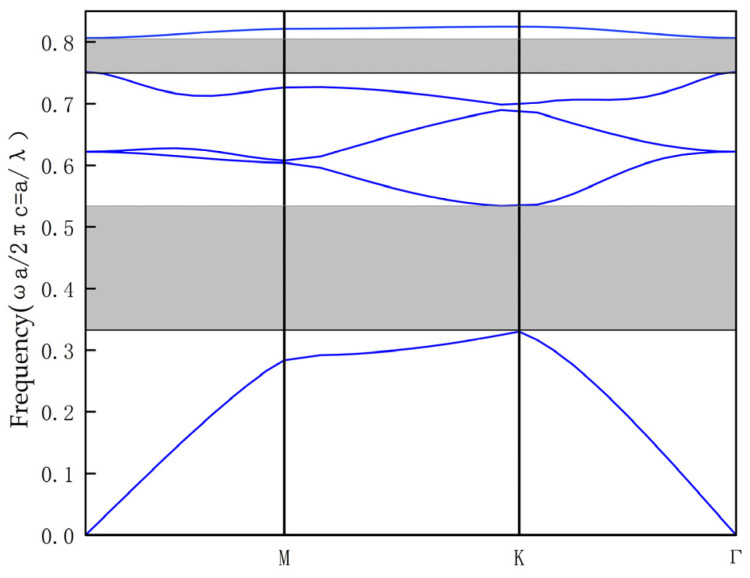
Band diagram for circular rods in 27 × 21 triangular lattice without any defects.

**Figure 2 micromachines-14-01478-f002:**
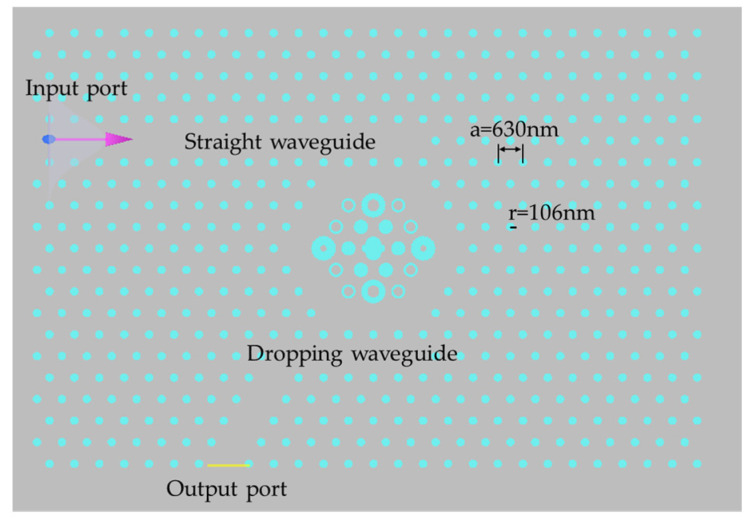
2D PC sensor.

**Figure 3 micromachines-14-01478-f003:**
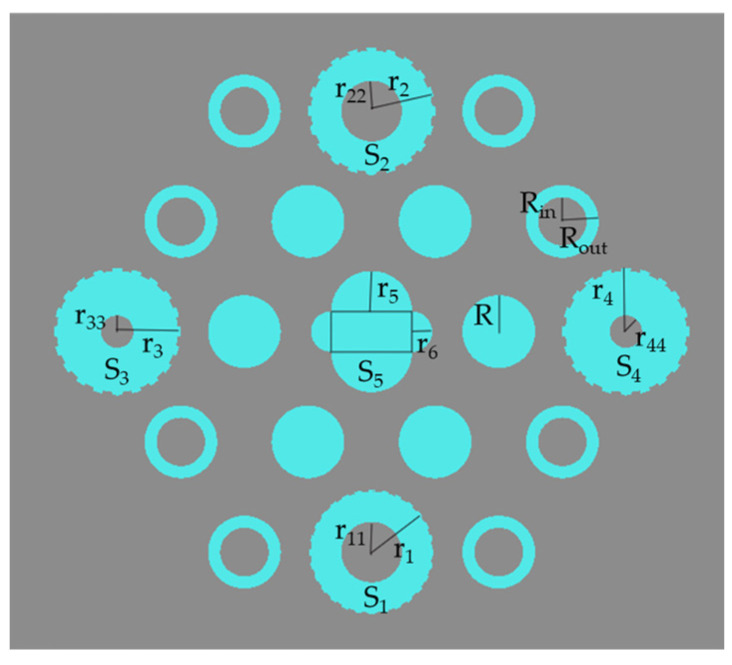
Amplified schematic diagram of the resonant cavity.

**Figure 4 micromachines-14-01478-f004:**
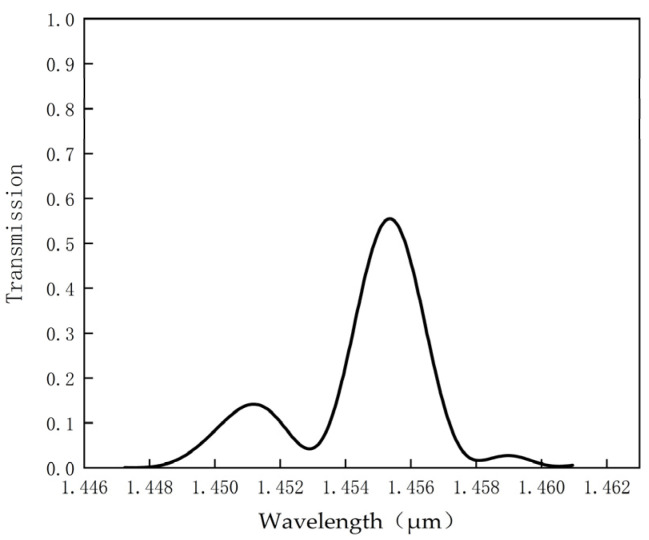
Transmission spectrum.

**Figure 5 micromachines-14-01478-f005:**
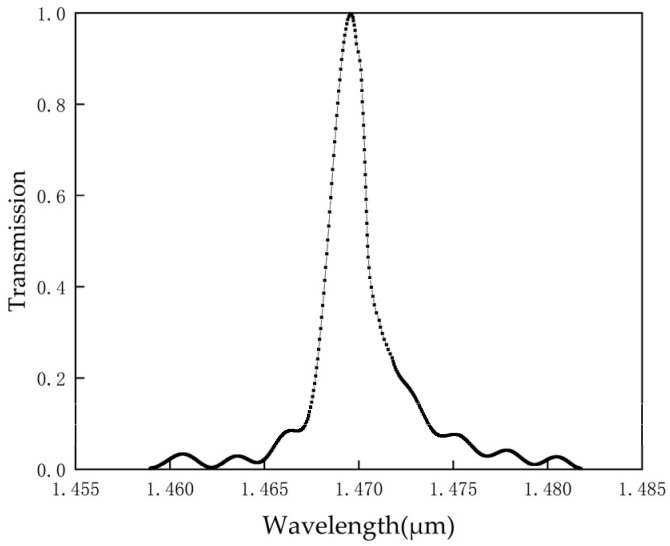
The transmission spectrum at the output monitor.

**Figure 6 micromachines-14-01478-f006:**
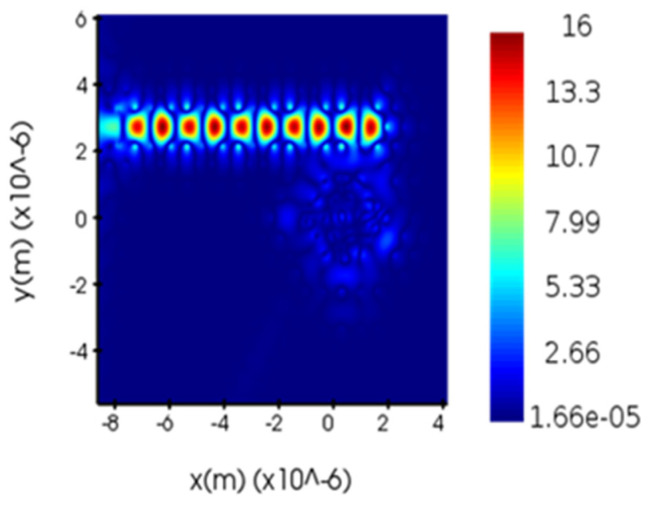
Optical power distribution at λ = 1455 nm.

**Figure 7 micromachines-14-01478-f007:**
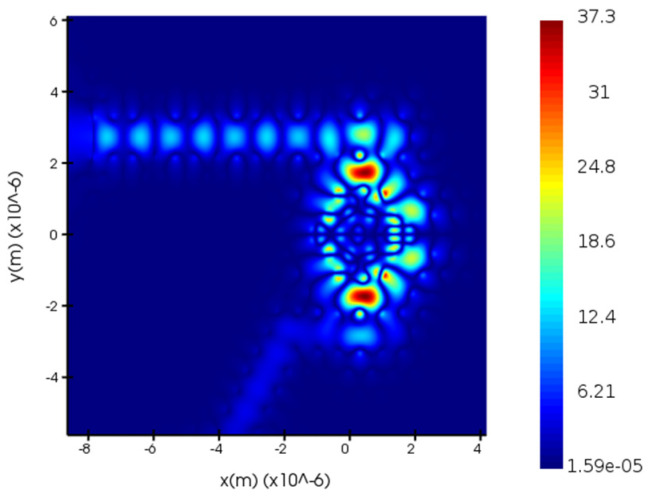
Optical power distribution at λ = 1469 nm.

**Figure 8 micromachines-14-01478-f008:**
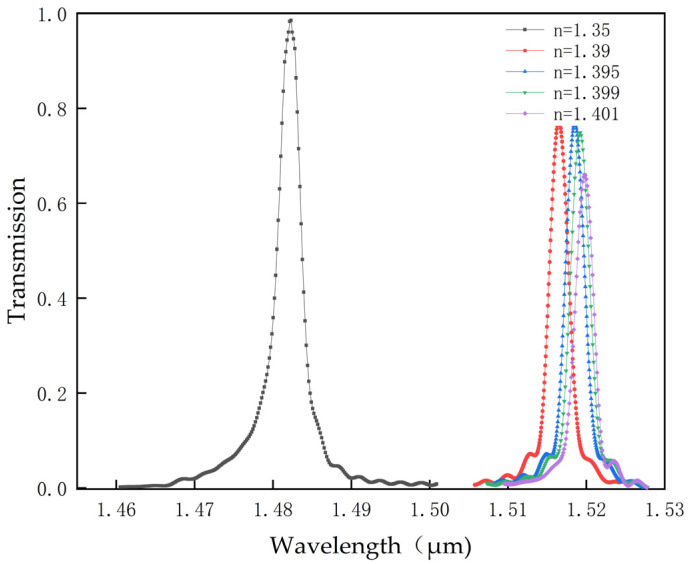
Transmission spectrum of the designed biosensor for different normal and cancer cells.

**Table 1 micromachines-14-01478-t001:** Steps taken to optimize the proposed structure.

r_1_ (um)	r_2_ (um)	r_3_ (um)	r_4_ (um)	Resonant Wavelength (nm)	*Q*	Transmittance (%)
0.2	0.3	0.3	0.3	1502.07	639	71.2
0.25	0.3	0.3	0.3	1455.29	537	55.8
0.3	0.3	0.3	0.3	1470.81	647	97.6
0.35	0.3	0.3	0.3	1493.28	496	88.26
0.4	0.3	0.3	0.3	1508.44	603	69.5
0.3	0.2	0.3	0.3	1524.16	554	47.8
0.3	0.25	0.3	0.3	1459.22	583	87.8
0.3	0.35	0.3	0.3	1493.55	551	99.96
0.3	0.4	0.3	0.3	1516.45	417	61
0.3	0.3	0.2	0.3	1491.76	438	93
0.3	0.3	0.25	0.3	1478.74	573	97
0.3	0.3	0.35	0.3	1488.78	633	98.74
0.3	0.3	0.4	0.3	1470.65	570	89.2
0.3	0.3	0.3	0.2	1490.15	545.84	88
0.3	0.3	0.3	0.25	1478.47	555.81	95.36
0.3	0.3	0.3	0.35	1503.24	538	55
0.3	0.3	0.3	0.4	1469.58	980	99.62

**Table 2 micromachines-14-01478-t002:** Parameters obtained from sensors using S_1_, S_2_, S_3_, and S_4_ intermediate columns as detection sources.

Detection Source	Refractive Index	Resonant Wavelength (nm)	*Q*	Wavelength Shift (nm)	*S* (nm/RIU)	*DL* (RIU)
S_1_ as the detection source.	1.35	1481	510	-	-	-
	1.39	1517.63	702	36.63	915.75	0.000236
	1.395	1518.95	534.84	37.95	843.3	0.000336
	1.399	1519.84	512	38.84	792	0.0003748
	1.401	1520.89	531	39.89	782.15	0.000366
S_2_ as the detection source.	1.35	1519.84	853	-	-	-
	1.39	1547.45	650.2	27.61	690.25	0.000344
	1.395	1548.09	586	28.25	627.8	0.000421
	1.399	1548.61	543	28.77	587.14	0.000485
	1.401	1549.3	564	29.46	577.64	0.000475
S_3_ as the detection source.	1.35	1457.65	681	-	-	-
	1.39	1469.87	639	12.22	305.5	0.000792
	1.395	1472.49	779	14.84	329.77	0.000573
	1.399	1474.49	708	16.84	343.63	0.00061
	1.401	1475.29	602	17.64	345.88	0.000708
S_4_ as the detection source.	1.35	1469.37	489	-	-	-
	1.39	1493.73	682	24.36	609	0.00036
	1.395	1495.03	695	25.66	570.22	0.00038
	1.399	1495.73	688	26.36	537.95	0.000404
	1.401	1496.05	715	26.68	523.13	0.0004

**Table 3 micromachines-14-01478-t003:** Comparison of the proposed structure with other biosensors in the literature.

References	Sample Detection	*Q*	*S* (nm/RIU)	Transmission Power (%)	*DL*
Baratye et al. [31]	Cancer cell	3803.55	308.5	98.78	-
Khan et al. [32]	Cancer cell	1200	227	-	-
Asuvarana et al. [33]	Cancer cell	573	4615	95	0.0013
Bindal et al. [34]	Cancer cell	650	850	70	-
This work	Cancer cell	980	915.75	99.62	0.000236

## Data Availability

The article includes the datasets that support the findings and conclusions of this study.

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
