# Peer review of "Design of Photonic Crystal Biosensors for Cancer Cell Detection"

_micromachines, 2023, doi:10.3390/mi14071478_

Round 1
Reviewer 1 Report
Using simulations, the authors have proposed a biosensor for cancer cell detection. Such proposed biosensor is based on a hexagonal resonant cavity introduced in the center of silicon photonic crystal. The maximum sensitivity of the proposed biosensor is 915,75 nm/RIU and their minimum detection limit is 0,000236 RIU.
Overall, the manuscript needs a lot of improvement in writing and English usage and the proposed biosensor should be compared with other approaches using photonic crystal resonators.
1. The manuscript needs a lot of improvement in the presentation and discussion of the results, and does not recommend its acceptance for publication in Micromachines.
2. The novelty of the proposed biosensor should be discussed (deeply and in detail) in the manuscript by using some up-to-date references in the introduction section.
3. In the third section (Result Design and Simulation of Biosensors), the authors claimed that the, a hexagonal resonant cavity is constructed by y introducing point defects at the center of the PhC. However, various point-defect photonic crystal cavities have already been well proposed with high-quality factors and small mode volume (Kassa-Baghdouche, L., & Cassan, E. (2020). Optical and Quantum Electronics, 52(5), 260, Kassa-Baghdouche, L. (2020). JOSA B, 37(11), A277-A284). These point defect PhC cavities should be discussed in the manuscript. Moreover, the difference between these point defect PhC cavities and the proposed hexagonal resonant cavity should be included in the manuscript.
4. In the third section (Result Design and Simulation of Biosensors), the authors used only the quality factor, sensitivity and detection limit for analyzing the characteristics of the proposed biosensor. However, there are many criteria for analysis the characteristics of the proposed biosensor such as : the shift of the resonant wavelength and the mode volume. Such criteria’s should be also used for analyzing the characteristics of the proposed biosensor.
5. In the manuscript, the authors mentioned that the quality factor Q of the proposed resonant cavity is about 980. Compared with point-defect photonic crystal cavities, the quality factor of the proposed resonant cavity is small (Kassa-Baghdouche, L. (2019). Physica Scripta, 95(1), 015502, Kassa-Baghdouche, L., Boumaza, T., Cassan, E., & Bouchemat, M. (2015). Optik, 126(22), 3467-3471.). We can enhance the detection limit with this designed resonant cavity?
6. The discussion needs to be presented in a more clear manner, presenting a summary or results and their relation to published literature. The authors do not clearly present their conclusions and more text should be dedicated to this section.
7. In this manuscript, only numerical simulations were performed by the authors for analyzing the characteristics of the proposed biosensor and any experimental results were not given. If the numerical results of this study have been extended with additional experimental results, it would be better.
English usage in this manuscript must be substantially improved. There are many grammatical errors and vague descriptions.
Author Response
Thanks for your review,for your guidance on this article, please take a look at the response letter.

Reviewer 2 Report
The authors' paper provided an interesting and potentially beneficial notion.
However, the writers should address a few concerns, as well as make a few technical changes.
In terms of technical corrections, the authors' photos are not of acceptable quality, both in terms of resolution and in terms of marking the sizes on the axes. It appears that the authors did not devote enough time and attention to these aspects. It is vital to develop graphics with sufficient quality and labels with sufficient clarity so that every reader understands what it is about. Furthermore, fonts and font sizes must be consistent throughout all photos. For physical quantities or mathematical notations, conventional notations must also be used. Despite this, the images in the description lack explanations. It is not acceptable to write solely as in Figure 1. Band structure.
Concerning the section 3 important corrections, the authors state:
"The band diagram of the PC structure is obtained by using the finite difference time domain method, as shown in Figure 1."
The method and procedure used to calculate the band structure for a specific PC should be presented in this section. In the same section, the optimization of the said structure is explained, which seems to have been done on the basis of trial and error. Is there a physical explanation for the mentioned optimization, that is, a physical explanation for the choice of the mentioned structure parameters?
Author Response

(The authors gave the same response as above.)

Round 2
Reviewer 1 Report
The authors have implemented my suggestions. I recommend this manuscript should be accepted for publication.
Minor editing of English language required
Reviewer 2 Report
The authors responded to the suggestions and I propose that the paper be accepted.